# Characterization of Cold Tolerance of Immature Stages of Small Hive Beetle (SHB) *Aethina tumida* Murray (Coleoptera: Nitidulidae)

**DOI:** 10.3390/insects12050459

**Published:** 2021-05-16

**Authors:** Muhammad Noor-ul-Ane, Chuleui Jung

**Affiliations:** 1Agricultural Science & Technology Research Institute, Andong National University, Andong 36744, Korea; mnoor493@hotmail.com; 2Department of Plant Medicals, Andong National University, Andong 36744, Korea

**Keywords:** invasive species, supercooling point, acclimation, and honey bee pest

## Abstract

**Simple Summary:**

Establishment and distribution of invasive insects depends on their cold tolerance especially in temperate regions. The small hive beetle (SHB) is native to Sub-Saharan African countries, from where it has now invaded all over the world, including South Korea as a destructive pest of honey bees. Therefore, the present study first time provided information about the survival and adaptation capacity of immature stages of SHBs to tolerate the cold environment. All tested immature stages: feeding larvae, wandering larvae and pupae of SHB were sensitive to chilling injuries. However, wandering larvae and pupae showed substantially enhanced survival in cold lethal temperatures after acclimation, but not the feeding larval stage. This plasticity of cold tolerance in wandering larvae and pupae could contribute to the winter survival of the SHB population for better establishment and range expansion in temperate regions such as in Korea.

**Abstract:**

The small hive beetle (SHB) *Aethina tumida* Murray, (Coleoptera: Nitidulidae) is now a global invasive pest of honey bees, but its cold tolerance potential has not been yet explored. Therefore, we measured the supercooling point (SCP) of different stages of SHBs and also the impact of acclimation on their SCPs and survival as a measure for cold tolerance. Combinations of different temperatures (0, 3, 5, 7, and 10 °C) for different hours (1, 3, 5, 7, 12, 24, 35, and 48 h) were used to assess SHB survival. The supercooling points occurred at lower temperatures (−19.4 °C) in wandering larvae than in the other stages (pupae: −12.5 °C, and feeding larvae: −10.7 °C). A lethal temperature (LT_50_) of feeding larvae was achieved earlier at 4.9 °C after 7 h exposure than the wandering larvae (3.7 °C at 48 h) and pupae (5.6 °C at 48 h). The sum of injurious temperature (SIT) is the most suitable estimation to describe cold resistance of the SHB immatures. The wandering larvae were the most cold tolerant, followed by pupae and feeding larvae based on SIT values of −286.8, −153.7 and −28.7 DD, respectively, and also showed more phenotypic plasticity after acclimation than feeding larvae and slightly more than pupae. Our results show that all stages, i.e., feeding larvae, wandering larvae and pupae, are chill susceptible. However, these stages, especially wandering larvae and pupae, showed the capacity to acclimate to cold temperatures, which may help them to survive in winter for the continuity of the SHB population, especially in a scenario of climate change.

## 1. Introduction

Temperature is one of most influential factors for an insect’s development and reproduction [1,2]. Thus, it affects the survival and distribution of insects in general. Winter temperatures cause substantial fitness challenges to insects for their survival, particularly in temperate regions [3,4]. A major threat for insects in temperate regions is freezing at sub-zero temperatures. Insects mainly use two strategies to survive cold weather: 1. freeze tolerance, i.e., the ability to survive ice formation in the body fluid, and 2. freeze avoidance, i.e., the ability to keep body fluids liquid by suppressing their supercooling points (SCP = temperature at which body fluids freeze). Insects that do not have either of these surviving strategies at subzero temperatures and die well above their SCPs are called chill susceptible insects. Therefore, a study of the relationship between SCPs and mortality should help to shed light on a species’ cold tolerance potential.

Phenotypic plasticity of insects is also used to determine thermal limits in addition to inherent genetic mechanisms. Acclimation is an adaptive phenotypic plasticity [5], and according to the beneficial acclimation hypothesis [6,7], it depends on species and duration of exposure. Acclimation to lower sub-lethal temperatures for days to several weeks increases cold tolerance of insects by depressing their SCPs [8,9]. Phenotypic plasticity for cold tolerance can vary across different life stages [10]. An SCP may not always be identifiable as the lowest temperature threshold to survive under, especially if a species is freeze-intolerant [11]. Therefore, the duration of lower temperature exposure should be linked with the SCP to understand cold thermal limits of insects [12,13]. Temperature and time relationship could also be used to calculate the upper limit of the cold injury zone (ULCIZ), i.e., the temperature above which cold does not cause mortality even after an ecologically meaningful period of time, and the sum of injurious temperature (SIT), temperature with long exposure that cause 50% mortality [14].

The small hive beetle (SHB), *Aethina tumida* Murray (Coleoptera: Nitidulidae) is an invasive pest of honey bee (*Apis mellifera*), stingless bees and bumble bees throughout the world [15,16,17], including South Korea [18]. Sub-Saharan African countries are the native habitat of SHB, where the species is a conditional pest of honey bee colonies [19,20]. The honey bee industry of South Korea is under threat from SHB populations that were first reported from *Apis mellifera* colonies in Miryang City in 2016 [21]. Once established inside a hive, the beetles can cause reductions in colony performance and hive production [22]. SHB larvae feed on honey brood, stored pollen food and honey, and their adults feed preferentially on pollen and honey in the hive. The adults may also carry pathogens such as *Paenibacillus larvae*, which cause American foulbrood disease in honey bees [23].

As feeding larvae live in honey bee hives, they usually do not experience cold temperatures as compared to other stages. However, there are still possibilities when SHB feeding larvae can be exposed to lethal cold temperatures. In our previous study [24], we reported that SHB can continue reproduction at the temperature of winter bee’s cluster (20 °C), so feeding larvae still can be exposed to winter temperatures in the brood-less area or peripheral portion of the hives. Colony loss, especially near winter season, may force SHB adults to lay eggs outside the hives near rotten fruits, which are alternative food sources [25] for their feeding larvae. Therefore, feeding larvae can potentially be exposed to lethal cold temperatures in that case too. After three to six days of feeding, third instar larvae stop feeding and move out of the hive as wandering larvae, regardless of the season, to find a suitable substrate (mostly soil) for pupation. Wandering larvae make pupal chambers after finding substrate and turned into pupae after spending 10 days in a pupal chamber at 25 °C [26]. The pupae take eight days in that substrate to become adults at 25 °C.

Being ectothermic, temperature greatly influences the development and survival of SHB in Korea, as we demonstrated in our previous study [26]. Winter survival is an important predictor of an invasive species to become established and to extend its range limits in temperate regions [3]. Additionally, SCP and winter mortality can be used in mechanistic models to understand possible invasion limits of the SHB [27]. Therefore, cold tolerance parameters should be explored in this invasive beetle. For this, we determined SCPs of SHB immature stages (feeding larvae, wandering larvae and pupae), their cold tolerance strategies, lethal temperatures causing 50% mortality over a constant period of time (LT_50_), ULCIZ, SIT, and role of acclimation to tolerate freezing injury and in their phenotypic plasticity.

## 2. Materials and Methods

### 2.1. Source of Insect

Adult SHBs were hand-picked from an apiary in Miryang City (35°48’ N, 128°75’ E), Gyeongsangnam-do, South Korea in 2016 and maintained in acrylic cages (38 × 38 × 34 cm) at Andong National University in a rearing room at 25 ± 2 °C, 60 ± 5% RH, and 12:12 h L/D photoperiod. Sand sterilized by autoclaving and then moistened (20% volume/weight) was placed on the bottom (7 cm depth) of the cage for pupation. Pollen patty-honey (pollen dough) was provided as a diet for the adults and larvae [18]. Pollen patty diet was composed of rape seed pollen, sugar, soybean flour, yeast and nutrient supplement in 20:60:20:20:1 ratio by mass. SHB adults were also provided 10% honey–water solution by volume soaked in cotton. SHB completed all of its life stages in the acrylic cages.

### 2.2. Supercooling Point

We determined SCPs of 1-d-old feeding larvae, 3-d-old wandering larvae, and 1-d-old pupae that were taken from the above-mentioned laboratory culture [9]. Twenty individuals of all SHB stages were used to determine their SCP. All SHB stages were attached with type-T copper thermocouple (BTM-4208SD, LT Lutron, Taipei, Taiwan) placed individually into 1.5 mL microcentrifuge tubes. Temperature was recorded at 1 s intervals. The thermocouple was attached with transparent adhesive tape to the thorax of all SHB stages except the pupae. For the pupae, we made two holes in the PCR tube and tied the thermocouple with copper wire passing through the holes so that the pupae were in contact the thermocouple tip by glycerin gel. Four PCR tubes containing individual SHBs were put in a styrofoam box (30 × 30 × 15 cm), which was then placed in a refrigerator at −40 °C until the thermocouples reached a temperature of −30 °C. The cooling rate was measures as 0.6 °C/min.

### 2.3. Cold Tolerance Strategy

Cold tolerance strategies of SHB stages were determined in separate sets of 12 individuals of each SHB stage per treatment. We placed SHB stages individually in PCR tubes and cooled the latter as described above for the SCPs. We moved all frozen or unfrozen individuals from Styrofoam (after observing exotherm in half of the individuals) to laboratory at 25 °C with 1 g of pollen patty as a food for feeding larvae in plastic bottle (20 mL). Wandering larvae and pupae were also separately placed in plastic bottles, containing moist sand (20% by *v*/*w*). The survival was scored for coordinated movement of each mobile SHB stage after 24 h. Pupae survival was scored for its successful emergence as adults from the pupae.

### 2.4. Lethal Temperature Determination

Seven individuals of each SHB stage were placed in clear plastic vials (20 mL). Pollen patty (1 g) diet was provided in case of feeding larvae. These plastic vials were then placed in an incubator (ET.IR-250.260707, Daihan Scientific, Wonju, Korea) for 1, 3, 5, 7, 12, 24, 36 and 48 h at temperatures range of 0, 3, 5, 7 and 10 °C. We stopped exposing SHB to temperatures when we got 100% mortality. Plastic vials containing fresh diet for feeding larvae, and moist sand (20% by *v*/*w*) for wandering larvae and pupae were transferred to laboratory at 25 °C. Survival was scored as described above. The experiment was replicated five times.

### 2.5. Acclimation

To determine SCP plasticity of SHB immature stages, feeding larvae were acclimatized at 5 °C for 5 h, and 10 °C for 1 d and 16 °C for 5 d, wandering larvae and pupae were acclimatized at 5 °C for 1 d, 10 °C for 2 d and 16 °C for 5 d. After acclimatization, their SCPs and survival were recorded after following the same procedure as mentioned in cold tolerance strategy. Twenty individuals were used for each stage to determine SCP. Twelve individuals per immature stage per treatment were used to check transformation of cold tolerance strategy.

To determine phenotypic plasticity, all SHB immature stages, were acclimatized for 5 days at 16 °C and then exposed to their respective LT_80_ values based on the above mentioned lethal temperature experiment. The survival of these stages was also scored as mentioned above. Seven individuals per replication were used for survival, and the experiment was replicated five times.

### 2.6. Statistical Analysis

The difference between SCP of SHB immature stages, effect of temperature on SHB mortality of each stage at its respective lethal hours and acclimation response in relation to SCP was measured by one-way analysis of variance (ANOVA), and means were compared by Least Significant Difference (LCD) test at *p* ≤ 0.05. Two-way ANOVA was applied to examine the interaction of temperature and its duration on SHB stages. Student *t*-test was applied to determine difference between acclimation and control for each SHB stage. Arcsine–square root transformation was performed for each percentage data before analysis. All statistics were done by using SPSS v20 software. Probit analysis was used to calculate the LT_50_ and LT_80_ for each life stage by using Polo Plus 2.0, LeOra Software. The parameters of Equations (1) and (2) were estimated by Statistica software (version 13).

The survival of SHB immature stages that is a function of time (*t*), and temperature (*T*) of SHB immature exposure assessed by following Equation (1) [14]:(1)$$S\;\left(t,T\right)=\frac{e^{a+bt\left(T-c\right)}}{1+e^{a+bt\left(T-c\right)}}$$
where *a*, *b*, and *c* are key parameters describing the time–temperature–survival relationship during cold exposure.

When *S* = 0.5, *SIT* (Summation of injurious temperature) would be following (2) [14];
(2)$$SIT=\frac{-a}{b}=\left(T-c\right)t$$

*a* and *b* are same parameters as above, and *c* = ULCIZ.

## 3. Results

### 3.1. SCP and Cold Tolerance Strategy

Under controlled conditions, the SCPs of different SHB stages varied significantly (F_2,57_ = 64.5: *p* < 0.001) (Table 1). The SCPs value of −19.4 °C of the wandering larvae was the lowest we recorded. No significant difference was apparent between the SCP of pupae (−12.5 °C) and feeding larvae (−10.7 °C). No single immature SHB stage survived ice formation inside their body, thereby exhibiting a lack of freeze tolerance (Table 2).

### 3.2. Lethal Temperature Determination

Environmental temperatures significantly affected the survival of feeding larvae (*F*_4,160_ = 315.1: *p* < 0.001), wandering larvae (*F*_4,160_ = 358.3: *p* < 0.001), and pupae (*F*_4,160_ = 200.0; *p* < 0.001). Similarly, survival of all stages significantly varied in relation to exposure hours (feeding larvae (*F*_7,160_ = 2791.4: *p* < 0.001), wandering larvae (*F*_7,16_ = 549.2: *p* < 0.001), and pupae (*F*_7,160_ = 360.5: *p* < 0.001)). Temperature and exposure interactions also varied among the SHB stages (feeding larvae: *F*_28,160_ = 90.12: *p* < 0.001, wandering larvae: *F*_28,160_ = 101.8: *p* < 0.001, and pupae: *F*_28,160_ = 33.0: *p* < 0.001).

All feeding larvae died when exposed for 7 h at 0 °C, whereas all wandering larvae and pupae died when exposed for 48 h at 0 °C (Figure 1). LT_50_ values were achieved quickly in feeding larvae during 7 h at 4.9 °C (Table 3). LT_50_ and LT_80_ of wandering larvae were achieved at 3.7 and 2.5 °C, respectively, when exposed for 48 h. Pupae were the second most cold tolerant stage along with feeding larvae based on their LC_50_ and LC_80_ values at 5.6 and 3.7 °C, respectively, on exposure duration of 48 h.

The model (Equation (1)) explained 99, 98, and 96% of total variations of the feeding larvae, wandering larvae and pupae, respectively (Table 4), showing good fit for the survival of immature stages of SHB. Estimated ULCIZ were not different among the immature stages (feeding larvae: 9.0 °C, wandering larvae: 10.1 °C and pupae: 9.1 °C). However, estimated SIT (Equation (2)) of wandering was −286.8 DD, which was higher in feeding larvae and pupae SIT values (−28.7 and −153.7 DD, respectively).

### 3.3. Acclimation

Acclimation did not change SCPs of all SHB stages at the tested temperatures (feeding larvae: *F*_3,76_ = 1.9: *p* = 0.136, wandering larvae: *F*_3,76_ = 1.93: *p* = 0.131, pupae: *F*_3,76_ = 1.38: *p* = 0.256), indicating that acclimation did not improve the ability to survive against freezing injury. Acclimation significantly increased (*t* = 2.9: *p* < 0.05) survival of feeding larvae over control. Similarly, wandering larvae and pupae significantly (*t* = 12.8: *p* < 0.001, *t* = 7.3: *p* < 0.001, respectively) increase survival over control (Figure 2).

## 4. Discussion

SCP of the feeding larvae in the present study is not consistent with what was observed in pollen beetle, *Meligethes aeneus* (Fabricius), belonging to the same Nitidulidae family [28]. Pollen beetle larvae have a lower supercooling point of −21 °C than feeding larvae of SHB in the present study. This difference could be related to a higher concentration of efficient nucleators [29] in pollen beetle larvae. In the present study, the higher supercooling point of feeding larvae than wandering larvae, which is a non-feeding stage, might be due to the presence of food in the alimentary canal. The ice nucleation due to food in the gut may start earlier at warmer temperatures, which decrease the supercooling capacity of the feeding larvae [30].

Since freezing killed all immature SHBs, it suggested that the SHB is freeze intolerant, just like *Meligethes aeneus*, which also belongs to the Nitidulidae [28]. Furthermore, temperatures above the SCPs also killed all immatures, indicating that all SHB stages are chill susceptible like other tropical originated insects, e.g., *Drosophila melanogaster* [31]. Therefore, it was important to see whether acclimation can help them to tolerate freezing. Acclimation did not affect the SCP of immature stages of SHB. No improvement of SCP through acclimation is also evident from other chill susceptible insects [31,32]. These results show that a considerable population of the SHB immature stages is vulnerable to a decline during winter time in temperate regions.

Temperature and exposure duration significantly affect the survival of the immature stages, and given the low SCP in wandering larvae, the latter can therefore be expected to survive colder conditions better than the other developmental stages. Survival of insects against low temperatures is highly dependent on intensity of temperature and exposure duration, with mortality increased with decreasing temperature and/or increasing exposure period [33]. Being chill susceptible, high mortality was expected in SHB immature stages at temperatures well above the SCP. Our results show that SCP is not a representative of the minimum lethal temperatures of immature stages of SHB.

Mortalities increased sharply at lower temperatures and reduced significantly with slightly higher temperatures. This mortality pattern is also shown in a previous study [34], and another beetle, *Entomoscelis Americana*, at lower temperatures [35], which also pupate in the soil. ULCIZ values are almost similar, ranging from 9–10 °C, among all tested SHB stages, which shows that above this range no chilling mortality will occur. However, there was no survival at 13 °C based on our previous developmental study [26]. This failure of development may link to the incomplete blockade of morphogenetic processes. This small window of quiescent state (>10 °C to <13 °C) of SHB immature stages also represents their chill susceptible nature. Wandering larvae as the cold resistant stage followed by pupae and feeding larvae based on the combination of survival, temperature and time (SIT). These results suggest that SIT is a more suitable indicator to differentiate cold hardiness of SHB immature stages.

Acclimation did not cause change in the SCP of any tested immature stage, which shows that SHB acclimation did not support beneficial acclimation hypothesis (BAH), i.e., “acclimation supports an organism to perform better in a particular environment in terms of SCPs. The same relation of acclimation and SCP is also presented in the case of another chill susceptible insect, *D*. *melanogaster* [31].

Acclimation marginally increased the survival of feeding larvae, which suggests its limited phenotypic plasticity against the cold temperatures. The feeding larvae would not survive the winter season inside the hive if honey bees are absent in the hives due to winter mortality, and outside the hives where temperature is freezing in temperate regions. In winter, honey bees maintain their nesting temperature around 15–20 °C in the form of cluster [36]. SHB larvae would take advantage of the warm cluster of winter honey bees. Phenotypic plasticity was more widespread in wandering larvae and pupae. This plasticity may help them to adapt to cold weather as the temperature starts to decrease during late fall.

The present study further strengthens the speculation of SHB’s survival in colder areas [34] by its phenotypic plasticity, though low to moderate winter mortality can still be expected in temperate regions. These results also support the prediction of future range expansion of SHB towards North Africa and some parts of Europe [37]. The existence of SHB in Canada [38] may also be attributed to SHB’s adaptive capacity against cold temperatures, but its range expansion still could be limited to certain climatic conditions in Canada [38]. This phenotypic plasticity can help to survive winter in areas having mean monthly temperature ranges 0 to −2 °C during winter in South Korea. However, this plasticity of SHB can lead to more survival under field conditions because of temperature fluctuation in the field. For example, larval and pupal stages of a chrysomelid beetle, *Entomoscelis americana* managed to survive at low temperatures by using thermo-regulation to cope with fluctuating and detrimental temperatures of the field [35]. Wandering larvae can move towards the area where soil is more covered by leaf debris, which can further reduce cold intensity to the wandering larvae and pupae. Wandering larvae can go deep, 20 cm into soil [39], where the temperature would not be equal to outside environmental temperatures, which will further increase survival of wandering larvae and pupae. Wandering larvae make pupal chambers when they enter into soil, which may make their surrounding temperature warmer than the actual soil temperature. The snow cover increases the soil temperature because of its low thermal conductivity and resulted in varied soil temperatures across the regions in South Korea [40]. Thus, areas under more snow depth will provide further protection to wandering larvae and pupae in combination with their phenotypic plasticity. Heat island effects of cooler cities may also help for a sustainable population buildup of the species such as SHB, which are sensitive to lower temperatures. High survival of SHB’s wandering larvae and pupae, even with a small increase along with its plasticity, may also help them to survive constraining temperatures, especially with a moderate warming scenario (RCP2.6) as applied in a previous study [34], which also predicted range expansion of SHB with global warming. In South Korea, winter temperature is increasing recently in association with the Arctic Oscillation [41], which may help further expansion of SHB in South Korea toward the northern region. This winter warming would help the survival of SHB in the mild cold regions of South Korea, where mean monthly temperature ranges from −2 to −3 °C during winter.

## 5. Conclusions

In conclusion, the present study shows that the SCP of wandering larvae and pupae (non-feeding stages) was lower than feeding larvae (a feeding stage). All SHB’s immature stages (feeding larvae, wandering larvae and pupae) are susceptible to chilling injury. Survival of immature stages decreased with the increase in temperature duration and intensity. These temperature durations and intensities can be incorporated in the development of correlative and mechanistic models of SHB [31]. SIT is the best indicator to differentiate cold resistance among SHB immature stages and suggested that wandering larvae can tolerate cold better than feeding larvae and pupae. The SCP is not a representation of lower lethal temperature of any tested SHB stage. However, the acclimation exhibited phenotypic plasticity across the different life stages of SHB. The wandering larvae may get acclimated in the winter bees cluster to survive winter outside the hive. This phenotypic plasticity may lead to increased cold tolerance among the immature stages through the evolutionary process. To survive winter, these stages may need alternate warm places, e.g., greenhouse in which greenhouse substrates can be alternate options for wandering larvae and pupae [42]. Wandering larvae and pupae may get additional favor from natural fluctuated temperatures in combination with other possible temperature increasing factors such as debris covered soil, snow depth, urbanization, global warming to survive and establish in temperate regions such as South Korea. The adaptive capacity of SHB stages against cold stress shows a need to alert Integrated Pest Management (IPM) specialists to monitor SHBs even during winter, which otherwise can profoundly contribute to their establishment in spring season. Our data suggests that predictive models for distribution and expansion for SHBs can also take help from plasticity of these immature stages for cold hardening response of SHBs. This study raises the concern that these stages with cold plasticity may find some refugia for population reservoir where these can avoid harsh environment.

## Figures and Tables

**Figure 1 insects-12-00459-f001:**
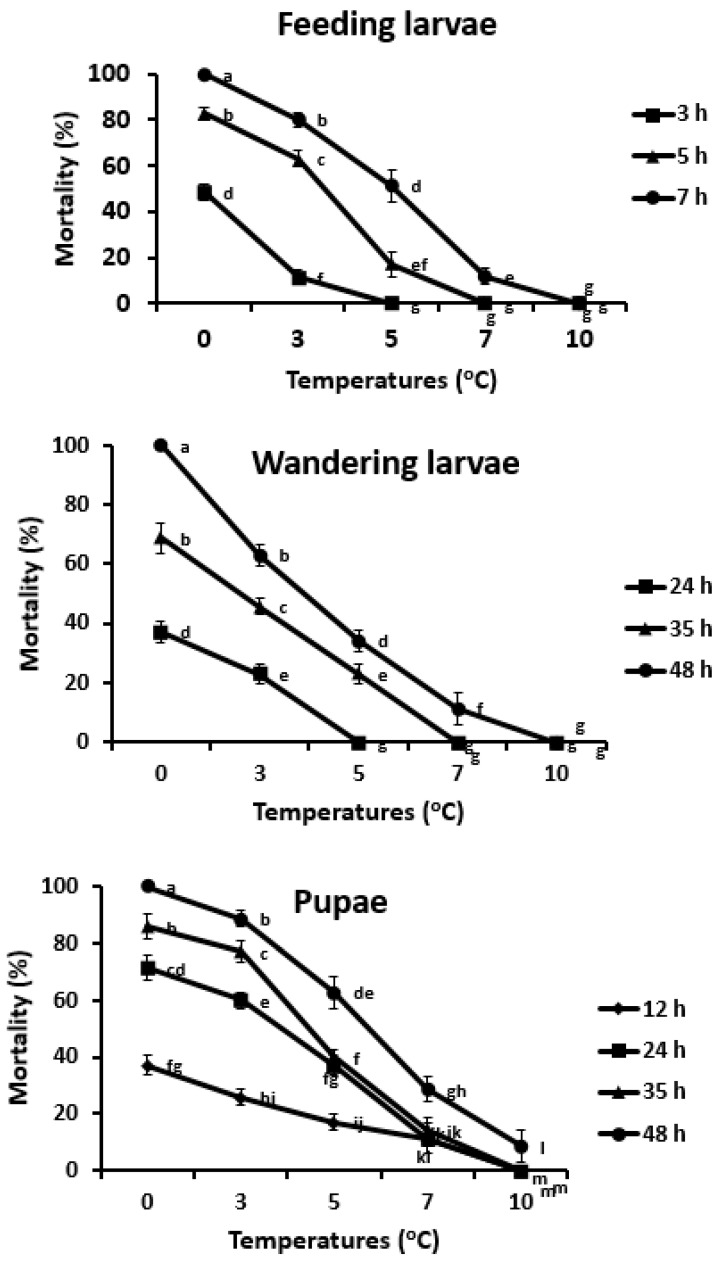
Percent mortality of feeding larvae, wandering larvae and pupae of SHB at different temperatures with lethal hours only (with respective to each stage).

**Figure 2 insects-12-00459-f002:**
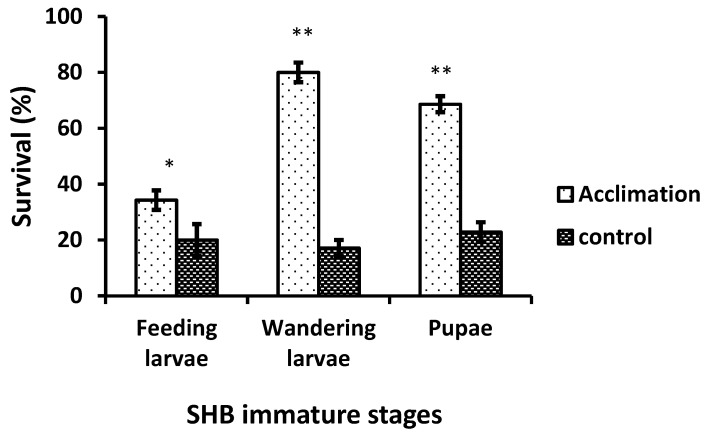
Survival of immature stages of SHB after acclimation. * *p* < 0.05, and ** *p* < 0.001.

**Table 1 insects-12-00459-t001:** Stage-specific supercooling points of SHB (± standard error) with acclimation and without acclimation.

Stage	Without Acclimation	Acclimation		
		16 °C	10 °C	5 °C
Feeding larvae	−10.7 ± 0.62c a *	−9.7 ± 0.48 a *	−9.6 ± 0.53 a *	−8.9 ± 0.44 a *
Wandering larvae	−19.4 ± 0.35a a *	−18.2 ± 0.40 a *	−18.4 ± 0.48 a *	−18.5 ± 0.32 a *
Pupae	−12.5 ± 0.69b a *	−10.9 ± 0.75 a *	−11.9 ± 0.68 a *	−12.2 ± 0.50 a *

* Comparison of means for acclimation treatments. Means sharing the same letter are not significantly different via Turkey’s HSD test.

**Table 2 insects-12-00459-t002:** Mortality of immature stages of SHB with acclimation and without acclimation to determine cold tolerance strategy.

Stage	Without Acclimation		Acclimation					
			16 °C		10 °C		5 °C	
	Mortality		Mortality		Mortality		Mortality	
	Unfrozen	Frozen	Unfrozen	Frozen	Unfrozen	Frozen	Unfrozen	Frozen
* Feeding larvae	12/12	12/12	12/12	12/12	12/12	12/12	12/12	12/12
* Wandering larvae	12/12	12/12	12/12	12/12	12/12	12/12	12/12	12/12
* Pupae	12/12	12/12	12/12	12/12	12/12	12/12	12/12	12/12

* Dead/total.

**Table 3 insects-12-00459-t003:** Lethal temperature to kill 50% and 80% population (LT_50_ & LT_80_) of each SHB stage at lethal hours (0–100% mortality).

Stage	Lethal Hours			
	7		48	
	LT_50_(°C) (95% CI)	LT_80_(°C) (95% CI)	LT_50_(°C) (95% CI)	LT_80_(°C) (95% CI)
Feeding larvae	4.9 (4.43–5.28)	3.5 (3.02–3.92)	**	**
Wandering larvae	*	*	3.7 (3.221–4.18)	2.5 (1.85–2.91)
Pupae	*	*	5.6(5.05–6.22)	3.7(3.02–4.18)

* Cannot be calculated because of zero mortality. ** Experiment was not performed further as 100% mortality was already observed at early hours.

**Table 4 insects-12-00459-t004:** Parameter estimates (±SE) of survival model (Eq. 1) of immature stages of SHB.

Stage	*a **	*b*	*c*	*R* ^2^
Feeding larvae	4.6 ± 0.64	0.16 ± 0.03	9.1 ± 0.56	0.99
Wandering larvae	4.1 ± 0.53	0.01 ± 0.01	10.1 ± 1.04	0.98
Pupae	2.7 ± 0.39	0.01 ± 0.00	9.1 ± 1.03	0.96

* *a*, *b* and *c* are the estimated parameters.

## Data Availability

All relevant data is provided is present in MS.

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
