# Peer review of "Characterization of Cold Tolerance of Immature Stages of Small Hive Beetle (SHB) Aethina tumida Murray (Coleoptera: Nitidulidae)"

_insects, 2021, doi:10.3390/insects12050459_

Round 1

Reviewer 1 Report

General comment:

The temperature in the hives could be higher than outdoors. Therefore, the experiments on cold resistance are relevant only for the life stages that occur outside the hives. Please, indicate how often feeding larvae occur outside the hives.

Or, if feeding larvae rare occur outside the hives, please indicate why the results obtained for them are also important.

Specific comments:

Line 3 Please, do not use hyphens in the text, especially in the title of the article.

Line 10  invaded insects replace by invasive insects

Line 15Please, replace semicolon to colon (immature stages:)

Lines 24 and 25 Are all stages (wandering larvae, pupae and feeding larvae) overwintering? The estimation of the supercooling point is important only for the overwintering stages.

Line 28-29 “the coldest tolerant stage” – it is not clear

Line 46 chill-susceptible – The word is broken in two lines.

Lines 75-76 – Please, explain if the wandering larvae can move out of the hive in winter. Do they go out of the hive in any season?

Line 88 Please, explain the method of collection. As I understand, the beetles were collected outside the hives. Yes? What method was used?

Pitfall trap? Barrier trap? What baits were used? Or the beetles were collected by sweep-netting of grasses near the hives?

Line 89 Does the hole life cycle of the beetles: from adults to adults took place in the acrylic cages?

Lines 124-125 “When we achieve 100% mortality at any temperature, we did not expose SHBs further for any temperature/hour. until we achieved 100% mortality at any temperature.” It is not clear. Why “100% mortality at any temperature” is repeated?

 Line 244 “can a find host” change to “can find a host”

Author Response

The temperature in the hives could be higher than outdoors. Therefore, the experiments on cold resistance are relevant only for the life stages that occur outside the hives. Please, indicate how often feeding larvae occur outside the hives.

Or, if feeding larvae rare occur outside the hives, please indicate why the results obtained for them are also important.

Response: Thank you for your comment. Yes, feeding larvae rarely live outside the hives. However, there are still possible options when they can expose to winter temperatures e.g. SHB reproduction in winter cluster and colony loss near winter seasons. We explained these points in texts as, “As feeding larvae lives in honey bee hives, they usually do not experience cold temperatures as compared to other stages. However, there are still possibilities when SHB feeding larvae can exposed to lethal cold temperatures. In our previous study [24], we reported that SHB can continue reproduction at the temperature of winter bee’s cluster (20oC) so feeding larvae still can be exposed to winter temperatures in the brood less area or peripheral portion of the hives. Colony loss especially near winter season, may force SHB adults to lay eggs outside the hives near the rotten fruits which are alternative food sources [25] for their feeding larvae. Therefore, feeding larvae can potentially expose to lethal cold temperatures.  (L 74-86)

Specific comments:

Line 3 Please, do not use hyphens in the text, especially in the title of the article.

Response: Thank you for your comment. We deleted hyphens Revised as, “Characterization of cold tolerance of immature stages of small hive beetle (SHB) Aethina tumida Murray (Coleoptera: Nitidulidae)”. (L3, 173,191-196, 232-237)

Line 10  invaded insects replace by invasive insects

Response: Thank you for your comment. We replaced as suggested and the lines were revised as “Establishment and distribution of invasive insects depends on their cold tolerance especially in temperate regions.”  (L10-11)

Line 15Please, replace semicolon to colon (immature stages:)

Response: Thank you for your comment. We replaced semicolon with colon and lines were revised as, “all tested immature stages: feeding larvae, wandering larvae and pupae of SHB were very sensitive to chilling injuries”. (L 15)

Lines 24 and 25 Are all stages (wandering larvae, pupae and feeding larvae) overwintering? The estimation of the supercooling point is important only for the overwintering stages.

Response: Thank you for your comment. There is no report on overwintering of these wandering larvae, pupae and feeding larvae. However, SCP points can be checked for any stage of insects which can face temperature below 0oC and become freeze. Being temperate region, SCPs is worthy to study for any stage of insects which can expose frequently or rarely to the cold temperatures in South Korea.”

Line 28-29 “the coldest tolerant stage” – it is not clear

Response: Thank you for your comment. We made it clear by revising the sentence as the wandering larvae were the most cold tolerant followed by pupae and feeding larvae based on SIT values of -286.8, -153.7 and -28.7 DD, respectively. (L 29-30)

L 17: Line 46 chill-susceptible – The word is broken in two lines.

Response: Thank you for your comment. We corrected and revised the sentence as our results show that all stages i.e. feeding larvae, wandering larvae and pupae are chill susceptible.  (L 18)

Lines 75-76 – Please, explain if the wandering larvae can move out of the hive in winter. Do they go out of the hive in any season?

Response: Thank you for your valuable comment. Yes, wandering larvae always go out of hives to find out the suitable substrate (mostly soil) for the pupation regardless of the season. For clarification, we also revised the lines in text as, “After 3-6 days of feeding, third instar larvae stop feeding and move out of the hive as wandering larvae regardless of the season to find the suitable substrate (mostly soil) for pupation. Wandering larvae make pupal chamber after finding substrate and turned into pupae after spending 10 days in pupal chamber at 25oC [26]. The pupae take 8 days in that substrate to become adults at 25oC” (L 82-86)

Line 88 Please, explain the method of collection. As I understand, the beetles were collected outside the hives. Yes? What method was used?

Pitfall trap? Barrier trap? What baits were used? Or the beetles were collected by sweep-netting of grasses near the hives?

Response: Thank you for your comment. No, we collected SHB adults inside the hives and simply hand-picking was used. We clarified and revised sentence as, “Adult SHBs were hand-picked from an apiary in Miryang City (35°48'N, 128°75'"E), Gyeongsangnam-do, South Korea in 2016” (L 100)

Line 89 Does the hole life cycle of the beetles: from adults to adults took place in the acrylic cages?

Response: Thank you for your comment. Yes, all life cycle was carried out in acrylic cages. We added the sentence in the text as, “SHB completed all of its life stages in the acrylic cages”.  (L 108)

Lines 124-125 “When we achieve 100% mortality at any temperature, we did not expose SHBs further for any temperature/hour.” It is not clear. Why “100% mortality at any temperature” is repeated?

Response: Thank you for your comment. We revised line in text as, “We stopped exposing insects to -cold temperatures when we got 100% mortality”. (L137-138)

Reviewer 2 Report

The last paragraph of the introduction outlines good reasons for this study into temperature tolerance of a pest insect and whether acclimation could help pest insects establish in a new area and even extend their range. However the results are not discussed in this context. The discussion and conclusion do not put the results into an applied context of the potential for this pest to acclimate and become a big problem in colder areas. Previous  research would suggest that SHB would have trouble becoming a major pest in cold areas this manuscript could challenge that from the acclimation findings. The discussion and conclusion need to apply the results to the SHB lifecycle in the environment of an apiary. Currently the discussion and conclusion appear too theoretical and of little use to anyone in beehives and concerned about the potential of this insect. There needs to be more discussion of the significance of the results in terms of previous research findings as well as the potential of this pest under the environmental conditions in Korea.

The discussion and conclusion are also a bit repetitive. The finding that wandering larvae and pupae are slightly more cold tolerant that feeding larvae is logical as these two stages live outside of the temperature controlled hive. 

Figure 1 top graph does not have a complete legend, and "7 is omitted from the text (line 181-182) where Figure 1 is referenced. Figure 2 is not referenced in the text at all. The results descriptions are a little confusing. 

Author Response

Reviewer #2: Dear authors,

The last paragraph of the introduction outlines good reasons for this study into temperature tolerance of a pest insect and whether acclimation could help pest insects establish in a new area and even extend their range.

Response: Thank you for your comment. We also intended to enlarge our knowledgebase on this scope and determined the possible population traits which could help SHB invasiveness in temperate regions.

 However the results are not discussed in this context. The discussion and conclusion do not put the results into an applied context of the potential for this pest to acclimate and become a big problem in colder areas. Previous research would suggest that SHB would have trouble becoming a major pest in cold areas this manuscript could challenge that from the acclimation findings. The discussion and conclusion need to apply the results to the SHB lifecycle in the environment of an apiary.

Response: Thank you for your comment. Very important points. We also exemplified the cases in Canada and also in Northern USA where SHB had become problematic even in cold area. Also discussion was expanded the possible expansion in Korea too.

Currently the discussion and conclusion appear too theoretical and of little use to anyone in beehives and concerned about the potential of this insect. There needs to be more discussion of the significance of the results in terms of previous research findings as well as the potential of this pest under the environmental conditions in Korea.

Response:  Thank you for your valuable comment. This clearly improved our MS significantly. The discussion and conclusion is revised as suggested. (L265-267, L282-321).

-Lines 124-125 “When we achieve 100% mortality at any temperature, we did not expose SHBs further for any temperature/hour.” It is not clear. Why “100% mortality at any temperature” is repeated?

Response: Thank you for your comment. We revised this in text as, “We stopped exposing SHB to temperatures when we got 100% mortality”. (L37-138)

The discussion and conclusion are also a bit repetitive. The finding that wandering larvae and pupae are slightly more cold tolerant that feeding larvae is logical as these two stages live outside of the temperature controlled hive.

Response: Thank you for your comment. We revised discussion and conclusion to remove repetition. 

Figure 1 top graph does not have a complete legend, and "7 is omitted from the text (line 181-182) where Figure 1 is referenced. Figure 2 is not referenced in the text at all. The results descriptions are a little confusing. 

Response: Thank you for your comment. Fig. 1 is drawn only for lethal hours (where we got mortality) of each stage. E.g. when we exposed stages to at 10, 7, 5, 3 and 0oC for 1 hour duration and if the mortality was 0% at any of those temperatures, that hour was not plotted. If we got at list 1% mortality, then we called that hour as a lethal hour. Lines 181-182 were deleted for better understanding of the results. Fig. 2 is mentioned (L 234). The results are revised now for clarity (L199-204). 

Round 2

Reviewer 1 Report

The authors have answered all questions and improved the text according the recommendations.